# Investigating the Role of TGF-β Signaling Pathways in Human Corneal Endothelial Cell Primary Culture

**DOI:** 10.3390/cells12121624

**Published:** 2023-06-14

**Authors:** Inès Aouimeur, Tomy Sagnial, Louise Coulomb, Corantin Maurin, Justin Thomas, Pierre Forestier, Sandrine Ninotta, Chantal Perrache, Fabien Forest, Philippe Gain, Gilles Thuret, Zhiguo He

**Affiliations:** 1Laboratory of Biology, Engineering and Imaging for Ophthalmology (BiiO), EA2521, Faculty of Medicine, Jean Monnet University, 42270 Saint-Etienne, France; 2Eye Bank, Etablissement Français du Sang (EFS) Auvergne-Rhône-Alpes, 42023 Saint-Etienne, France; 3Ophthalmology Department, University Hospital Center, 42055 Saint-Etienne, France

**Keywords:** corneal transplantation, corneal endothelial cells (CECs), TGF-β, endothelial–mesenchymal transition (EndMT), endothelial cell density (ECD), cell therapy, tissue-engineered endothelial keratoplasty (TEEK)

## Abstract

Corneal endothelial diseases are the leading cause of corneal transplantation. The global shortage of donor corneas has resulted in the investigation of alternative methods, such as cell therapy and tissue-engineered endothelial keratoplasty (TEEK), using primary cultures of human corneal endothelial cells (hCECs). The main challenge is optimizing the hCEC culture process to increase the endothelial cell density (ECD) and overall yield while preventing endothelial–mesenchymal transition (EndMT). Fetal bovine serum (FBS) is necessary for hCEC expansion but contains TGF-βs, which have been shown to be detrimental to hCECs. Therefore, we investigated various TGF-β signaling pathways using inhibitors to improve hCEC culture. Initially, we confirmed that TGF-β1, 2, and 3 induced EndMT on confluent hCECs without FBS. Using this TGF-β-induced EndMT model, we validated NCAM as a reliable biomarker to assess EndMT. We then demonstrated that, in a culture medium containing 8% FBS for hCEC expansion, TGF-β1 and 3, but not 2, significantly reduced the ECD and caused EndMT. TGF-β receptor inhibition had an anti-EndMT effect. Inhibition of the ROCK pathway, notably that of the P38 MAPK pathway, increased the ECD, while inhibition of the ERK pathway decreased the ECD. In conclusion, the presence of TGF-β1 and 3 in 8% FBS leads to a reduction in ECD and induces EndMT. The use of SB431542 or LY2109761 may prevent EndMT, while Y27632 or Ripasudil, and SB203580 or SB202190, can increase the ECD.

## 1. Introduction

The human corneal endothelium consists of a single layer of 300,000 to 500,000 (depending on age and individual factors) corneal endothelial cells (CECs) covering the inner surface of the cornea. These cells maintain corneal transparency through their ion pump functions. In vivo, most human CECs (hCECs) are arrested at the G1 phase of the cell cycle [1,2]. In response to various diseases or damage, hCECs rely on cell migration and enlargement for repair [2,3]. Since the regeneration capacity of hCECs is limited in vivo, ex vivo, and in vitro, the quality of endothelial tissue is primarily evaluated using the endothelial cell density (ECD), measured in cells/mm^2^, in patients as well as in stored corneas and cultured cells.

Corneal endothelial diseases are common and are dominated by three main causes: a primary disease of the endothelium called Fuchs endothelial corneal dystrophy (FECD), an iatrogenic disease called pseudophakic bullous dystrophy (PKB), and the limited survival of some corneal grafts. FECD affects between 4% and 11% of adults over the age of 40 in the western world, to varying degrees [4]. PKB is a rare complication (between 0.1% and 0.5%) of lens surgery, but this is the most common operation worldwide, with an estimated 25 million procedures per year [5]. A corneal graft is the most frequent transplantation globally, with around 185,000 procedures per year [6]. The number of hCECs in the graft decreases progressively in all recipients, but the rate varies significantly from person to person. In some cases (the proportion of which is poorly estimated), the endothelium eventually becomes deficient. These three diseases share two points in common: the definitive loss of endothelial function associated with low ECD results in permanent corneal edema, and a corneal graft (also called keratoplasty) is the only available treatment to date.

The global shortage of corneal donations is a significant issue [6], highlighting the need for alternative therapies to replace donor corneas. One promising approach is hCEC culture-based therapy, which has two different modalities: injecting a suspension of hCECs into the anterior chamber [7] and tissue-engineered endothelial keratoplasty (TEEK) [8,9,10]. The large-scale dissemination of these methods relies on obtaining high yields of production of clinical-grade hCECs in vitro, which is particularly challenging [11]. Two major difficulties are the loss of differentiation that accompanies endothelial–mesenchymal transition (EndMT) [12,13,14] and low ECD (<<2000 cells/mm^2^) [13,15,16,17,18] due to the limited proliferation capacity of adult hCECs.

The in vitro proliferation of hCECs requires the use of fetal bovine serum (FBS) [17]. Regardless of the culture method employed, the culture medium typically contains 5–8% FBS to stimulate cell expansion. FBS contains approximately 10–20 ng/mL of TGF-β [19], which is known to initiate and maintain epithelial–mesenchymal transition (EMT) [20,21] and inhibits the G1/S transition of the cell cycle [22,23] in epithelial cells. In hCECs, TGF-β has also been shown to prevent S-phase entry [24,25,26], and inhibiting TGF-β receptors may suppress EndMT [27,28].

In the present study, we investigated the impact of the three isoforms of TGF-β and their signaling pathway inhibition on in-vitro-cultured hCECs, with the goal of providing recommendations to improve hCEC culture. In addition, we introduced a simple immunofluorescence (IF) method to evaluate the impact of a large number of molecules (inhibitors) on the quality of hCEC culture. Our approach involved first evaluating the effects of TGF-β1, 2, and 3 on confluent hCECs incubated in serum-free medium (SFM) to induce EndMT. Once confirmed, we assessed the sensitivity of six IF biomarkers using this TGF-β-induced EndMT model and identified NCAM as a reliable biomarker for further study. Since the in vitro expansion of hCECs requires the presence of FBS, we tested the effects of TGF-β1, 2, 3, and their neutralizing antibody on hCECs in a medium containing 8% FBS. Finally, we evaluated the inhibition of different TGF-β signaling pathways.

## 2. Materials and Methods

### 2.1. Ethical Statement

The hCECs used in this study were selected from our primary cultures of hCECs that were isolated from human corneas preserved in organ culture [29]. The human corneas were handled following the principles set forth in the Declaration of Helsinki. They were obtained from two sources: (i) cornea donation for research, authorized by the French Biomedicine Agency (PFS16-010)—we were authorized to select donors who presented contraindications for clinical use, and our research activity had no impact on donor cornea availability for transplantation; (ii) corneas discarded from the St. Etienne, Besançon, and Nantes eye banks because they were not suitable for transplantation.

### 2.2. Crucial Products

Culture media. *1—Descemet’s membrane digestion medium:* OptiMEM-I (11058021, Gibco^TM^, Grand Island, NY) supplemented with 20 µg/mL ascorbic acid (A5960, Sigma-Aldrich, St. Louis, MO, USA), 200 µg/mL CaCl_2_ (C5670, Sigma-Aldrich), and 2 mg/mL collagenase A (10103586001, Roche, Basel, Switzerland). *2—Serum-free medium (SFM):* Human endothelial SFM (11111-044, Gibco^TM^), antibiotic–antimycotic (15240062, Gibco^TM^) diluted to 1/200, CaCl_2_ at 200µg/mL, human serum albumin (HAS) (HGTH-250, GLOBAL^®^ TOTAL^®^) at 100 µg/mL, GlutaMAX (35050038, Gibco^TM^) diluted at 1/300, Insulin-Transferrin-Selenium (41400045, Gibco^TM^) diluted at 1/300, ascorbic acid at 20 µg/mL, FGF2 (SRP4037, Sigma-Aldrich) at 2 ng/mL, EGF (E5036, Sigma-Aldrich) at 2 ng/mL, PDGF BB (SRP3138, Sigma-Aldrich) at 2 ng/mL, β-NGF (#5221, Cell Signaling Technologies, Danvers, MA, USA) at 2 ng/mL. *3—Basic growth medium (BGM)*: OptiMEM-I, heat-inactivated FBS (CVFSVF00-01, Eurobio, Les Ulis, France) diluted to 8%, gentamicin (15710049, Gibco^TM^) at 50 µg/mL, CaCl_2_ at 200µg/mL, ascorbic acid at 20 µg/mL, EGF at 5 ng/mL, chondroitin sulfate (034-1462, FUJIFILM Wako Pure Chemical Corporation, San Diego, CA, USA) diluted at 0.08%. 4—*Complete growth medium*: BGM supplemented with 10 µM SB203580 (S1076, SelleckChem, Houston, TX, USA) and 1 µM SB431542 (S1067, SelleckChem).

TGF-βs and their neutralizing antibody. TGF-β1 (#78067, Stem Cell Technologies, Vancouver, BC, Canada), TGF-β2 (100-35B, PeproTech, Cranbury, NJ, USA), TGF-β3 (#243-B3, R&D Systems, Minneapolis, MN, USA), Fresolimumab (Hy-P99020, MedChemExpress, Monmouth Junction, NJ, USA), which is a neutralizing antibody against all the 3 isoforms of TGFβ. The concentration of TGFβ1, 2, and 3 in SFM or in BGM was 10 ng/mL and the concentration of the neutralizing antibody was 10 µg/mL.

Primary antibodies and secondary antibodies to reveal the 6 biomarkers. The three primary antibodies for the 3 biomarkers of CECs were anti-NCAM (MAB24081, R&D Systems), anti-CD166 (559260, BD Biosciences, San Jose, CA, USA), and anti-Na^+^/K^+^ ATPase (05-369, Sigma-Aldrich). The three primary antibodies for the 3 biomarkers of EndMT were anti-CD73 (Sc-32299, Santa Cruz Biotechnology), anti-Col 5A1 (Sc-20648, Santa Cruz Biotechnology, Dallas, TX, USA), and anti-TGFBI (10188-1-AP, ProteinTech, San Diego, CA, USA). The choice of these biomarkers is justified in the Discussion section. Two secondary antibodies were goat anti-mouse IgG (H + L) cross-adsorbed secondary antibody, Alexa Fluor™ 488 (A-11001, Invitrogen, Carlsbad, CA, USA) and goat anti-rabbit IgG (H + L) cross-adsorbed secondary antibody, Alexa Fluor™ 488 (A-11008, Invitrogen).

Inhibitors of the TGF-β signaling pathways. Inhibitors of six intracellular signaling pathways and TGF-beta receptors were tested; for each specific cell signaling pathway, we used two different inhibitors (Table 1).

### 2.3. Primary Culture of hCECs

The hCECs were obtained by primary culture, and the culture method is described in our previous study, which was a result of a collaboration with Professor Koizumi and the Okumura team [29]. Briefly, Descemet’s membrane, including the corneal endothelium, was mechanically peeled off from donor corneas, followed by incubation in 500 µL of Descemet’s membrane digestion medium at 37 °C for 16 h. After digestion, the released CECs were washed in OptiMEM-I using centrifugation at 200× *g* for 5 min. The cells were then seeded in a 24-well plate (190 mm^2^/well) coated with iMatrix-511 (892012, Nippi) and cultured in complete growth medium containing 8% FBS. The culture plate was kept in an incubator with a humidified 5% CO_2_ atmosphere at 37 °C, and the medium was replaced with fresh culture medium once per week until confluence was reached (passage (P)0). Subsequent passaging of the cells was performed at a ratio of 1:2, and the time from cell seeding to subculturing was approximately one month. We used seven hCEC cultures from seven different donor corneas that were preserved using the organ culture storage method. The donors’ ages ranged from 58 to 82 years, with an average of 72 ± 10 years (four females and three males). The initial ECD was 2624 ± 487 (1920–3248) cells/mm^2^ in the donor corneas. These seven cultures were selected based on their typical morphologies of hCECs, with an ECD higher than 1200 cells/mm^2^.

### 2.4. Immunofluorescence (IF)

The hCECs were seeded in 384-well plates pre-coated with iMatrix-511 at a ratio of 500 cells/mm^2^ and cultured in BGM alone (control) or BGM containing TGF-β1, 2, or 3, or one of the TGF-β cell signaling pathway inhibitors, for four weeks. The IF protocol was previously described by our team [29,42]. Briefly, cells were fixed in pure methanol at room temperature for 15 min after rinsing with PBS containing Ca^2+^ and Mg^2+^. The cells were then rehydrated in PBS and incubated in blocking buffer (PBS, 2% bovine serum albumin, 2% goat serum) for 30 min at 37 °C. The primary antibodies, diluted to 1/300 in blocking buffer, were incubated with cells at 37 °C for one hour under gentle agitation (30 rpm). After three rinses in PBS, the secondary antibodies, diluted to 1/600, and DAPI diluted at 2 µg/mL in blocking buffer were incubated with cells at 37 °C for one hour under gentle agitation. After three rinses in PBS, the cells were immersed in Fluoromount-G^TM^ mounting medium (00-4958-02, Invitrogen) to protect the fluorochromes (Alexa Fluor™ 488 and DAPI). An epifluorescence inverted microscope (IX81, Olympus, Tokyo, Japan) with the CellSens software (Soft Imaging System GmbH, Olympus) was used to acquire images.

### 2.5. Methods of Quantification

EndMT characterization by cell shape analysis

This analysis was performed based on phase-contrast images of hCECs obtained using a phase-contrast microscope (CKX41, Olympus) with a 10× objective. Due to the suboptimal quality of images from the 384-well culture plates (border effect), we used 24-well plates for cell culture and phase-contrast imaging. This analysis was performed only for the TGF-β-induced EndMT on confluent hCECs incubated with SFM (Result 1). Phase-contrast images of the cells were analyzed based on cell shape. Since typical hCECs possess a hexagonal or polygonal shape in vitro, while mesenchymal or fibroblast cells are elongated, we used the aspect ratio (AR) as the criterion to characterize EndMT. The AR was the ratio of the major axis to the minor axis of each cell. When the AR was close to 1, the cells were typical hCECs, whereas a higher AR indicated EndMT. The protocol for the AR measurement is detailed in Appendix A.

ECD measurement

To determine the ECD, we counted the number of cell nuclei stained with DAPI per square millimeter. All ECD counts were performed in 384-well plates pre-coated with iMatrix 511. To evaluate the effects of different molecules (TGF-β1, 2, or 3, neutralizing antibody anti-TGF-βs, and 14 TGF-β signaling pathway inhibitors) on cell expansion, hCECs were seeded and cultured in BGM (control) or BGM containing one of the molecules to be evaluated (Experiment/Result 3 and 4). The cultures were maintained for 4 weeks. An epifluorescence microscope equipped with a X10 objective was used to acquire one image per well, and the number of nuclei was counted on the entire surface of each image (1.6 mm^2^) using a home-made plugin installed in ImageJ. This microscopic field, representing 16% of the well surface, allowed the reliable measurement of ECD.

EndMT analysis by fluorescence intensity of biomarkers

Assessment of EndMT is crucial during hCEC culture and requires the use of an appropriate biomarker. In this study, we investigated EndMT via the fluorescence intensity of biomarkers in two steps. Firstly, we evaluated three specific biomarkers of hCECs (NCAM, CD166, and Na^+^/K^+^ ATPase) and three potential biomarkers of EndMT (CD73, TGFBI, and Col 5A1) using our TGFβ-induced EndMT model to select the most appropriate biomarker (Experiment/Result 2). We hypothesized that TGF-βs could decrease the fluorescence intensity of the three hCEC biomarkers and increase the fluorescence intensity of the three EndMT biomarkers. Secondly, we employed the selected biomarker (NCAM) to assess the effects of TGF-βs, their neutralizing antibody, and different TGF-β signaling pathway inhibitors on EndMT in hCECs that were cultured in the presence of FBS. All of these experiments were conducted using IF in 384-well plates. Alexa Fluor^TM^ 488 was used as the fluorescence tracker linked with the secondary antibody. Image acquisition was performed using an epifluorescence microscope with a FITC filter cube (Ex: 450–490 nm, Em: 500–550 nm, DM filter: 495 nm), with one image captured per well using the X10 objective. All parameters, including the intensity of the light source, exposure time, and image resolution, were consistently maintained throughout the experiments. Fluorescence control (IF using only the secondary antibody) was performed for each experiment. The mean gray value of each image was measured using Image J, and the staining intensity was calculated by subtracting the mean gray value of the fluorescence control from the mean gray value of the biomarker.

Data standardization for quantitative comparisons

In order to compare data across different cell culture replicates (obtained from different donors) and experiments (performed at different times), we standardized the data (measured in terms of AR, ECD, or fluorescence intensity) by comparing them to their own control groups (untreated hCECs). For example, in cell cultures A and B (from corneas of different donors), the ECD was 1000 and 1300 cells/mm^2^, respectively, in the control group (cells cultured in the medium without any evaluated molecules), and it was 650 and 900 cells/mm^2^, respectively, under TGF-β1 treatment. To enable comparison of these data between cultures A and B, we standardized them by calculating the ratio to their own controls. The standardized ECD of TGF-β1 was 0.65 for culture A (i.e., 650/1000) and 0.69 for culture B (i.e., 1000/1300).

Normalization of NCAM fluorescence intensity according to ECD

Since NCAM is present on the lateral membranes of CECs, the fluorescence intensity of NCAM staining may fluctuate with the ECD independently of EndMT. To address this, we normalized the signal on the cell perimeter, rather than solely on the ECD. To estimate the perimeters of hCECs from their ECDs, we assumed that hCECs were regular hexagonal cells. We calculated the perimeter from the ECD as follows: area of a regular hexagon (A) = (3√3 side length^2^)/2, perimeter = 6 × side length, and A = 1,000,000 µm^2^/ECD. Thus, the perimeter per cell (µm) was 6 √(2,000,000/(3√3 × ECD)). We calculated the fluorescence intensity per cell as follows: total fluorescence of the image = mean gray value x total pixels of image, total cell number in image = surface area (mm^2^) × ECD (cells/mm^2^). Thus, the fluorescence intensity per cell = (mean gray value × total pixels of image)/(surface of image × ECD). The normalized fluorescence intensity = fluorescence intensity per cell/perimeter per cell. Let us take TGF-β1-treated cells and control cells as an example. The mean gray value for TGF-β1 and the control is 100 and 200, respectively, with an ECD of 900 cells/mm^2^ and 1300 cells/mm^2^, respectively. The image resolution is 1024 × 1024 pixels, and its surface area is 1.6 mm^2^. To normalize the fluorescence intensity of a single cell by its perimeter, we used the formula (100 × 1024 × 1024/(1.6 × 900))/(6 √(2,000,000/(3√3 × 900))) for TGF-β1 and (200 × 1024 × 1024/(1.6 × 1300))/(6 √(2,000,000/(3√3 × 1300))) for the control. We calculated the ratio of the normalized fluorescence intensity of TGF-β1 to the control to standardize the data. This ratio is (100/200)/√(900/1300). The standardized fluorescence intensity is 100/200, and the standardized ECD is 900/1300. Therefore, the standardized normalized fluorescence intensity for TGF-β1 is the standardized fluorescence/√standardized ECD. The normalized fluorescence intensity was used in Experiments 3 and 4, where the ECD of some groups was significantly different from that of the control group.

### 2.6. Statistics

To compare the means of three or more independent groups and determine statistical significance, we used the one-way analysis of variance (ANOVA) test. When a significant difference was found, we performed Tukey’s honestly significant difference (HSD) post-hoc test to identify which treatments (TGF-βs or inhibitors) were significantly different from the controls and significant differences between any two groups. Graphs displayed the mean and standard deviation (SD) for each group. The data presented above each bar represented the mean ± SD (min, max). Statistical analysis and graph construction were performed using GraphPad Prism.

## 3. Results

### 3.1. The Three TGF-β Isforms Induced EndMT on Confluent hCECs in Absence of FBS

We first examined the impacts of the three TGF-β isoforms on confluent hCECs in the absence of FBS, since FBS contains various growth factors, including TGF-βs, as well as various hormones and biologically active substances that can potentially counteract the effects of added TGF-βs. The hCECs were cultured in complete growth medium for four weeks until full confluence in 24-well plates (for cell morphology analysis) and in 384-well plates (for ECD analysis). The cells were then exposed to SFM alone (control) or with 10 ng/mL of TGF-β1, 2, or 3 for seven days. To confirm that the observed biological effects were attributable to TGF-βs, we added 10 µg/mL of a neutralizing antibody against TGF-βs (1, 2, 3). The EndMT was characterized by analyzing the cell shape and quantified using the aspect ratio (AR) parameter, and the ECD was also assessed.

Cell elongation that corresponded to the induction of EndMT was consistently observed when TGF-βs were present, particularly TGF-β1 and 3. The neutralizing antibody against TGF-βs reversed these morphological changes (Figure 1A). Quantitative analysis confirmed the observation (Figure 1B). The AR of cells treated with TGF-β1, 2, or 3 was significantly higher than that of the control, and the AR of TGF-β1 and 3 was significantly higher than that of TGF-β2. The neutralizing antibody against TGF-βs significantly reversed or reduced the effects of TGF-βs. Although the neutralizing antibody could not completely reverse the effects of TGF-β3, a significant decrease was observed with TGF-β3 + neutralizing antibody compared to TGF-β3 alone. Exposure to TGF-βs or the neutralizing antibody did not significantly influence the ECD (Figure 1C)

### 3.2. Selection of NCAM as Biomarker to Assess EndMT

We used the TGF-β-induced EndMT model (in step 1) to identify a biomarker that could be used to quantify EndMT more easily and accurately. After the cells reached confluence, we induced EndMT in hCECs by incubating them in 384-well plates with SFM alone (control) or containing 10 ng/mL of TGF-β1, 2, or 3 for 7 days. We assessed each of the six preselected biomarkers (NCAM, CD166, Na^+^/K^+^ ATPase, CD73, TGFBI, and Col 5A1) using the IF technique. The images were acquired using the same parameters under a X10 objective during the same experiment (Figure 2A). The fluorescence intensity of each biomarker was then measured. A biomarker suitable for evaluating EndMT should exhibit a significant difference in fluorescence intensity between control and TGF-β-treated cells.

NCAM and Na^+^/K^+^ ATPase showed significant differences between TGF-β-treated cells and non-treated cells (control/SFM alone) (Figure 2C). NCAM was more sensitive than Na^+^/K^+^ ATPase due to the greater decrease in its mean intensity induced by TGF-βs. In addition, NCAM exhibited a significantly higher baseline fluorescence intensity compared to Na^+^/K^+^ ATPase (Figure 2B), which is of practical significance. When a marker displays a strong fluorescence signal, it relies less on the microscope’s performance, such as the camera sensitivity and fluorescence source power. This advantage of NCAM can facilitate the relative quantification of NCAM in cells treated with various substances. A series of IF images of NCAM on hCECs treated with TGF-β1, 2, or 3, with or without the neutralizing antibody is shown in Appendix A. The results confirmed the reliability of NCAM to highlight the induction of EndMT.

### 3.3. TGF-β1 and 3 Induced EndMT and Lower ECD on Cultured hCECs in Presence of FBS

In the primary culture of hCECs, FBS is necessary for cell expansion. To determine the impact of the three TGF-β isoforms on hCECs cultured in a basic growth medium (BGM) containing 8% FBS, we seeded hCECs at a density of 500 cells/mm^2^ in 384-well culture plates. The cells were cultured either with BGM alone as a control or with 10 ng/mL of TGF-β1, TGF-β2, or TGF-β3, in the presence or absence of 10 µg/mL of a neutralizing antibody against TGF-βs. After four weeks of culture, we evaluated the hCECs using IF, focusing on two main criteria, the ECD and EndMT, which was indicated by the fluorescence intensity of NCAM.

IF images were acquired using the X10 and X40 objectives and are shown in Figure 3A. In cells cultured in the BGM-based medium supplemented with 8% FBS, the addition of TGF-β1 or 3 led to a significant decrease in ECD compared to the control (BGM alone), while TGF-β2 had no significant effect. The addition of the neutralizing antibody increased the ECD significantly compared to the control (Figure 3B). Additionally, the fluorescence intensity of NCAM decreased upon the addition of TGF-β1 or 3, indicating the induction of EndMT (Figure 3C).

### 3.4. The Assessment of Inhibitors of TGF-β Signaling Pathways Revealed Positive Effects in Inhibiting P38-MAPK, ROCK, and TGF-β Receptor

One of the main goals of this study was to provide recommendations for the optimization of the primary culture of hCECs, which requires the presence of FBS. Therefore, we conducted this experiment using a culture medium containing FBS. We seeded hCECs at a density of 500 cells/mm^2^ in 384-well culture plates, using either BGM alone as a control or one of the two different inhibitors tested for each TGF-β signaling pathway. After four weeks of culture, we assessed the effects of TGF-β on the hCECs using immunofluorescence (IF), with a focus on two main criteria: the ECD and EndMT (NCAM fluorescence intensity). The cells were exposed to the different test molecules continuously, from seeding to confluence.

IF images illustrating the cell size and morphology, which is also a criterion used to assess cell quality, were acquired under a X40 objective and are shown in Figure 4A. Cells treated with both inhibitors of the Rho/Rock pathway (Y27632 and Ripasudil), P38 MAPK pathway (SB203580 and SB202191), and TGF-β receptor (SB431542 and LY2109761) showed a visible improvement in cell morphology by exhibiting more hexagonal and regular cell shapes. Quantitative results indicated that the ECD was significantly higher in cells treated with both inhibitors of the Rho/Rock and P38 MAPK pathways, whereas it was significantly lower with the two inhibitors of the Ras/Raf/MEK/ERK pathway (U0126 and AZD0364) (Figure 4B). Moreover, both inhibitors of the P38 MAPK pathway demonstrated a significantly higher ECD compared to the Rho/Rock pathway inhibitors. The anti-EndMT effect, reflected by the increase in NCAM fluorescence intensity, showed an improvement in cells treated with both inhibitors of the TGF-β receptor (Figure 4C). These results are summarized in Figure 5, which only considers when both inhibitors of the same signaling pathway showed corresponding and significant effects.

## 4. Discussion

### 4.1. Effects of TGF-β 1, 2, and 3 on In Vitro hCECs

Previous studies typically focused on only one of the three isoforms of TGF-β in investigating its effects on CECs, with TGF-β2 being the most commonly studied [26,43,44,45,46]. Our findings, however, demonstrate significant variations among the three isoforms, with TGF-β2 exhibiting a less severe negative impact on hCECs than the other two isoforms. This emphasizes the need to carefully consider the specific isoform being utilized in future studies.

The high-affinity human monoclonal antibody (Fresolimumab) that we utilized in our study functions to neutralize the active forms of human TGF-β1, 2, and 3 [47,48]. Nevertheless, the efficacy of this antibody against all three isoforms of TGF-β remains undetermined. Interestingly, our observations revealed that the neutralizing antibody effectively reduced the EndMT effect induced by TGF-β1 and 2, but only partially attenuated the EndMT effect induced by TGF-β3. This suggests that the neutralizing antibody may have variable efficacy against the different isoforms of TGF-β, with relatively lower efficacy against TGF-β3. In the absence of FBS, TGF-β1, 2, and 3 induced EndMT on confluent hCECs within one week, resulting in a decrease in NCAM’s fluorescence intensity of 60%, 36%, and 60% in confluent hCECs. In contrast, in the presence of FBS, TGF-β1, 2, and 3 induced EndMT within four weeks, leading to a decrease in NCAM’s fluorescence intensity of 40%, 22%, and 33% during cell expansion. Our hypothesis is that FBS may serve as a buffer against the effects of TGF-βs. FBS contains a variety of growth factors and hormones that are capable of modulating the TGF-β signaling pathways. Some of these factors may counteract the pro-EndMT effect of TGF-β. Additionally, FBS can provide essential nutrients and antioxidants that protect cells from stress-induced EndMT [49].

We confirmed that TGF-βs, especially TGF-β1 and 3, cause a significant decrease in ECD, which is another important criterion in evaluating the quality of in-vitro-cultured hCECs. This could be partially explained by the decrease in the proliferation of hCECs [22,23]. Eliminating TGF-βs using a neutralizing antibody from FBS can lead to a significant increase in ECD, providing further evidence of the harmful impact of TGF-βs on the ECD.

Conversely, beneficial effects of TGF-β have also been reported. When added to confluent cells cultured in a maturation medium, TGF-β1 can enhance the endothelial phenotype, in contrast to its induction of EndMT when added to cells grown in a proliferation medium [50]. Similarly, the improvement in the endothelial phenotype exerted by TGF-β2 in the same maturation medium was even greater than that of TGF-β1 [51]. We found some differences between the maturation medium and our medium without FBS. Specifically, the previous maturation medium contained FBS but lacked the addition of ascorbic acid (an antioxidant) and had a lower concentration of TGF-βs (2 ng/mL compared to our 10 ng/mL). As explained in the Discussion section, FBS may serve as a buffer against the pro-EndMT effect of TGF- βs. Furthermore, our ongoing study suggests that this buffering effect of FBS is more prominent in the ex vivo endothelium, where cells are in a confluent state. Moreover, low concentrations of TGF-β (1 ng/mL) are known to protect human trabecular meshwork cells from oxidative-stress-induced damage through the balance of p-AKT signaling [52]. One of the TGF-β signaling pathways, PI3K/AKT/mTOR, can play an antioxidant role [53,54], and it is known that oxidative stress is a critical factor in EMT engagement [49]. Referring to the trabecular meshwork cells, which are neighboring cells of hCECs, we speculate that low concentrations of TGF-βs could protect hCECs from EndMT through the antioxidant effect via the PI3K/AKT/mTOR pathway. Additionally, hCECs treated in maturation medium containing TGF-β at 2 ng/mL also showed stronger activation of the AKT pathway [51].

### 4.2. Proposal of a Screening System for Molecules to Evaluate Their Effects on hCECs In Vitro

Obtaining a large number of hCECs from primary cultures, particularly from the corneas of donors over 40 years of age, is a significant challenge since clinical-grade hCECs have been obtained to date only with donors below 30 years of age [7]. Screening various molecules, such as different growth factors, cell signaling pathway inhibitors, and coating molecules, can be helpful in identifying methods to optimize hCEC cultures. In this study, we present a simple method to screen molecules on hCECs cultured in 384-well plates using IF of NCAM and DAPI. Unlike phase-contrast observation, which remains difficult in small wells due to light reflection in wells’ walls, the small surface area of 10 mm^2^ per well in 384-well plates allowed us to minimize the quantity of cells without negatively impacting the IF observation. The two main criteria, the ECD and EndMT, can be easily and reliably assessed by counting the DAPI-stained nuclei and measuring the fluorescence intensity of NCAM. Counting and measurement can be performed using the free software ImageJ, which is a user-friendly tool for image analysis. Our proposed formula for the normalization of the NCAM fluorescence intensity to the ECD provides a reliable method to assess EndMT in hCECs. Moreover, the staining of NCAM alone can serve as visible evidence to assess the quality of CECs. The perfectly drawn morphology of CECs, as shown by NCAM, is also an important indicator in evaluating cell quality. Finally, it is important to note that having a common control is essential for each experiment and cell culture, as it allows for an accurate comparison between them.

The differentiation status of hCECs is rather complex to assess. It is known that hCECs readily undergo phenotypic transformation into fibroblasts by EndMT [14]. We therefore set up a simple method to quantitatively assess the differentiation and EndMT status of hCECs. Six biomarkers were firstly preselected. NCAM [55,56,57], CD166 [58,59], and Na^+^/K^+^ ATPase [11,58] are specific biomarkers for hCECs, and CD73 [59], TGFBI, and Col5A1 are potential biomarkers to assess EndMT. ZO-1 and N-cadherin, which are two other well-recognized, specific markers for hCECs [11,58], were also tested but were eliminated due to their low fluorescence intensity in cultured hCECs. SLC4A4 is frequently used as a specific biomarker for mRNA analysis [60,61], but its use in IF is still being discussed [58]. Vimentin, N-cadherin, collagen I, and fibronectin are well-known biomarkers for EMT but have not been validated for EndMT in hCECs. Vimentin and N-cadherin were ubiquitously expressed by normal hCECs [58], and the obvious immunostaining of collagen I and fibronectin was found in cultured hCECs without EndMT. CD73 is a fibroblast biomarker and has been used to assess EndMT status for hCECs [62], while Col 5A1 and TGFBI have been reported as markers of EMT [63,64,65]. The three biomarkers have been previously tested in our laboratory and have shown promising results, making them interesting candidates for this study. Among these six preselected markers (NCAM, CD166, Na^+^/K^+^ ATPase, CD73, TGFBI, and Col5A1), NCAM was chosen for its high fluorescence intensity in normal hCECs and high sensitivity to EndMT, making it the most suitable biomarker to quantify EndMT by measuring its fluorescence intensity. Moreover, NCAM is a lateral membrane marker that delineates the cell morphology, which is an additional criterion in characterizing the differentiation status of hCECs [58].

### 4.3. Effects of Inhibiting Various TGF-β Signaling Pathways on Cultured hCECs

While previous studies have reported the positive effects of individual inhibitors such as Y27632 (a ROCK inhibitor), SB203580 (a P38 MAPK inhibitor), and SB431542 (a TGF-β receptor inhibitor) on cultured CECs, a comparison of their effects is currently lacking. Furthermore, there is currently no research examining the effects of inhibiting different TGF-β cell signaling pathways. In this study, we targeted the six main intracellular signaling pathways [66,67,68,69,70] and the TGF-β receptor using their inhibitors, including Y27632, SB203580, and SB431542.

The ROCK inhibitor Y27632 is the most extensively studied molecule for hCECs and is considered as the critical factor behind the success of the first cell therapy to treat corneal endothelial diseases [7]. Its multiple biological benefits on CECs, promoting cell adherence, proliferation, and survival [71,72,73], have been reported. In this study, we observed that Y27632 had a dual positive effect on the ECD and anti-EndMT. The promotion of cell proliferation may explain its impact on the increase in the ECD. Observations of anti-EMT in epithelial cells [74,75] and the indication of anti-EndMT in hCECs [76] are consistent with our study. The positive effects of another ROCK inhibitor, Ripasudil (approved for clinical administration [77,78]), were inferior to those of Y27632.

The TGF-β receptor inhibitors, SB431542 and LY2109761, have both shown an anti-EndMT effect on hCECs. This anti-EndMT effect is consistent with the literature [27,28]. In addition to the anti-EndMT effect, LY2109761 improved the ECD, with a significant increase of 23% compared to the control. These two inhibitors are theoretically involved in the total inhibition of the TGF-β signaling pathway. The suppression of TGF-βs by their neutralizing antibody is also a method of total inhibition of the signaling pathway. It has shown a more beneficial effect on the ECD than on anti-EndMT. The non-completely specific effect of the inhibitors and the complexity of intracellular signaling pathways could be the causes. Compared to the two chemical inhibitors, the suppression of TGF-βs by their neutralizing antibody remains economically infeasible for the mass production of hCECs.

Both P38 MAPK inhibitors, SB203580 and SB202191, demonstrated excellent efficacy in increasing the ECD by 59% and 65%, respectively, while also maintaining the differentiated state of hCECs. SB203580 was reported for its protection of CECs’ barrier function in vitro and ex vivo [79,80] and its numerous beneficial effects on hCEC culture through anti-senescence mechanisms [38].

The two Inhibitors targeting the Ras/Raf/MEK/ERK pathways exhibited a negative effect on the ECD. These pathways are known to stimulate the growth of epidermal and epithelial cells, and our study highlights their importance for in vitro hCEC growth. This observation also demonstrates the complexity of the biological effects of TGF-β, which can promote the proliferation of certain cell types [81,82].

### 4.4. Limitations and Perspectives

In this study, we demonstrated that NCAM served as a reliable biomarker to assess EndMT induction due to its sensitivity, as evidenced by a significant decrease in fluorescence intensity. It is important to note that this decline in fluorescence intensity indicates a reduction in protein expression, which need be further confirmed through a series of Western blotting assays.

Because the number of primary hCECs was too small to perform both IF and Western blotting for each target, we chose to develop an IF-based analysis method in this study. We employed two methods to minimize potential errors: (1) we utilized two different inhibitors that targeted the same signaling pathway; (2) the selection of molecule concentrations was guided by the relevant literature, with priority given to studies involving CECs, as indicated by the references cited in Table 1.

In the field of corneal endothelia, there is an urgent need for a screening tool capable of evaluating a large number of molecules and optimizing the primary culture of hCECs. The proposed method aims to address this need. However, it is important to note that our approach has certain limitations that should be considered.

In this study, the effects of the various inhibitors were only observed in a single passage of the cells. To ensure their potential clinical application, it will be necessary to evaluate the stability of cells treated with these drugs/inhibitors over time, starting from the P0 culture passage.

Bonanno’s team suggested that lactate flux is a component of the corneal endothelial pump [83,84,85]. It would be highly intriguing to investigate whether there exists a correlation between endothelial lactate efflux function and EndMT in hCECs. The IF tool and TGF-β-induced EndMT models presented in this study could be used to conduct such a correlational study.

The team of Lee and Kay has extensively studied the effect of FGF-2 in EndMT on CECs in various publications [86,87,88,89]. It will be very interesting to exploit the different FGF-2 signaling pathways with the molecular screening tool set up in this study in order to identify other inhibitors that may have positive effects on CEC culture. In addition, other growth factors or cytokines known to have EMT effects, such as MCP-1 [90,91], TNF-α [92,93], and IL-1β [94,95], and their signaling pathways, could also be potential targets to study the optimization of the primary culture of hCECs.

## 5. Conclusions

In-vitro-cultured hCECs are susceptible to the negative effects of TGF-βs, particularly TGF-β1 and 3, which can induce EndMT and decrease the ECD. TGF-β receptor inhibition had an anti-EndMT effect. Inhibition of the ROCK pathway, notably that of the P38 MAPK pathway, increased the ECD, while inhibition of the ERK pathway decreased the ECD. For molecular screening studies, we recommend using the method presented in this study based on the IF of NCAM/DAPI.

## Figures and Tables

**Figure 1 cells-12-01624-f001:**
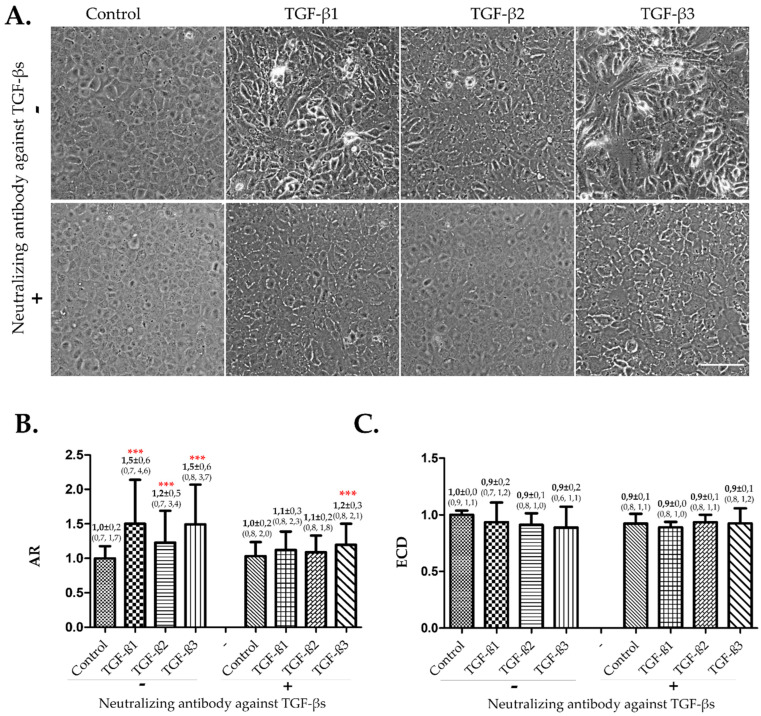
**Effects of TGF-β 1, 2, or 3 on confluent hCECs in the absence of FBS.** The confluent hCECs were incubated in SFM alone (control/-) or containing TGF-β 1, 2, or 3 in the absence or presence of a neutralizing antibody against TGF-βs. Incubation time was 7 days. (**A**) ***Illustration of cell morphology by phase-contrast images.*** The images were acquired using a phase-contrast microscope with X10 objective. Cropped images were used to illustrate cell morphology. Scale bar = 100 µm. (**B**) ***Quantitative comparison of EndMT using standardized AR.*** EndMT was assessed by cell morphology represented by the AR, which is the ratio of the major axis/minor axis of each cell. The AR data were obtained from manually tracing the contours of cells one by one from phase-contrast images (Appendix A). The higher the AR, the more severe the EndMT. *n* ≥ 87 cells for each group. (**C**) ***Comparison of the standardized ECD.*** ECD was assessed by nuclei staining using DAPI. *n* = 16 wells. At least two different cell cultures from two different donors were used for quantitative analysis of AR and ECD. Standardization of data was performed for each experiment and each cell culture by calculating the ratio with their own control/- (SFM alone without TGF-β, without neutralizing antibody). Only the significant differences from control/- were denoted by red stars on the corresponding bars: *** *p* < 0.0001.

**Figure 2 cells-12-01624-f002:**
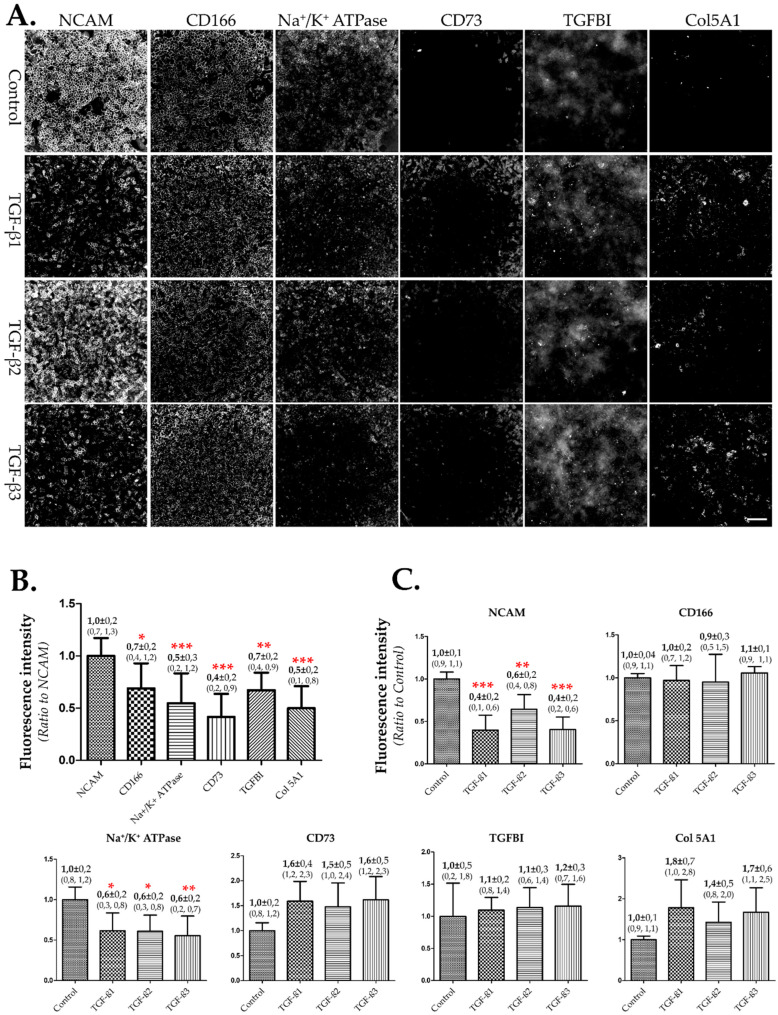
**Assessment of biomarkers using TGF-β-induced EndMT models.** After reaching confluence, EndMT was induced in confluent hCECs with TGF-β1, 2, or 3 in SFM for 7 days. The three CEC biomarkers (NCAM, CD166, and Na+/K+ ATPase) and three EndMT biomarkers (CD73, TGFBI, and Col 5A1) were evaluated in control cells (SFM alone) and in TGF-β-induced EndMT cells. (**A**) ***IF images of the 6 biomarkers.*** Images were obtained using an epifluorescence microscope with a FITC filter, using the same parameters for all images (intensity of light source, exposure time, contrast, resolution). The objective was X10, scale bar = 200 µm. (**B**) ***Fluorescence intensity of the 6 biomarkers in untreated cells (SFM alone).*** The fluorescence intensity of each biomarker was standardized by calculating the ratio with that of NCAM. Two different cultures from different donors were used (*n* = 12 wells). (**C**) ***Changes in fluorescence intensity under TGF-β 1, 2, or 3 for each of the 6 biomarkers.*** The sensitivity of each biomarker to EndMT of hCECs was evaluated by applying each of the 6 biomarkers with SFM alone (control) or containing TGF-β1, 2, or 3. The data were standardized by calculating the ratio with their own controls. *n* = 6 wells from 3 cell cultures were used for each group (control or TGF β1, 2, or 3) and for each biomarker. Significant differences by comparison to NCAM (**B**) and to controls (**C**) were marked with red stars on the corresponding bars: *** *p* < 0.0001, ** *p* < 0.001, * *p* < 0.005.

**Figure 3 cells-12-01624-f003:**
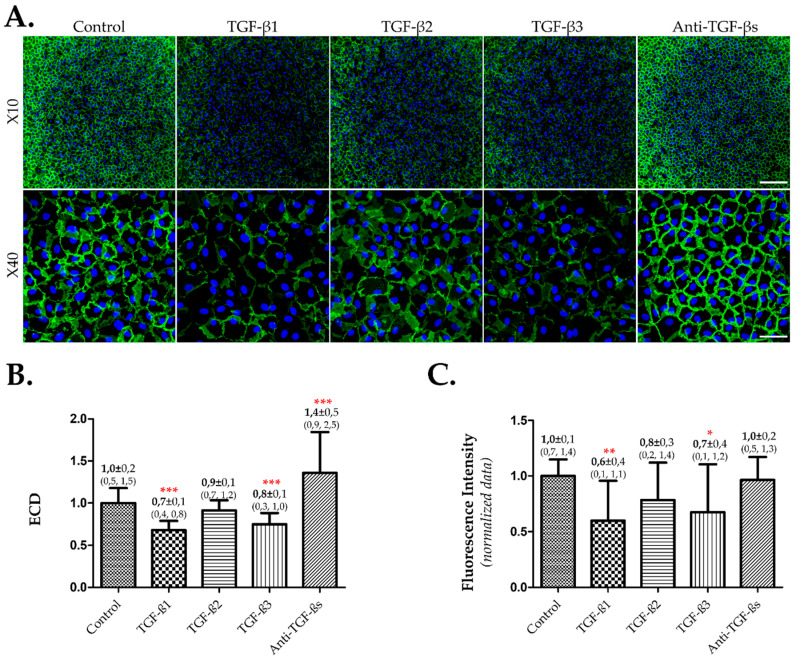
**Effects of TGF β 1, 2, 3 and neutralizing antibody anti-TGF-βs on hCECs cultivated in presence of FBS.** After cell seeding in 384-well plates, the hCECs were cultured in culture medium based on BGM that contained 8% FBS for 4 weeks. BGM only (control) or BGM containing 10 ng/mL of TGF-β1, 2, or 3 or 10 µg/mL of neutralizing antibody was applied. (**A**) ***Illustrative images.*** Cell lateral membranes were stained in green by NCAM and the nuclei were counter-stained in blue using DAPI. A higher magnification (X40) was added in order to show the cell morphology. The images were taken in the center of each well. Images of the same magnification were acquired with the same parameters. The scale bar was 200 µm for X10 images and 50 µm for X40 images. (**B**) ***Assessment of standardized ECD.*** ECD was assessed by nuclei counting using DAPI staining. The ECD data were standardized by calculating the ratio with controls. (**C**) ***Assessment of EndMT by normalized fluorescence intensity of NAM.*** The IF images of NCAM were acquired with an FITC filter using the same parameters. Normalized fluorescence intensity represents the fluorescence intensity per cell/cell perimeter as described in the Materials and Methods. The data were also standardized by calculating the ratio with the control. For B and C, at least two cultures from different donors were used (*n* ≥ 16 wells). Significant differences by comparison to controls were denoted by red stars on the corresponding bars: *** *p* < 0.0001, ** *p* < 0.001, * *p* < 0.005.

**Figure 4 cells-12-01624-f004:**
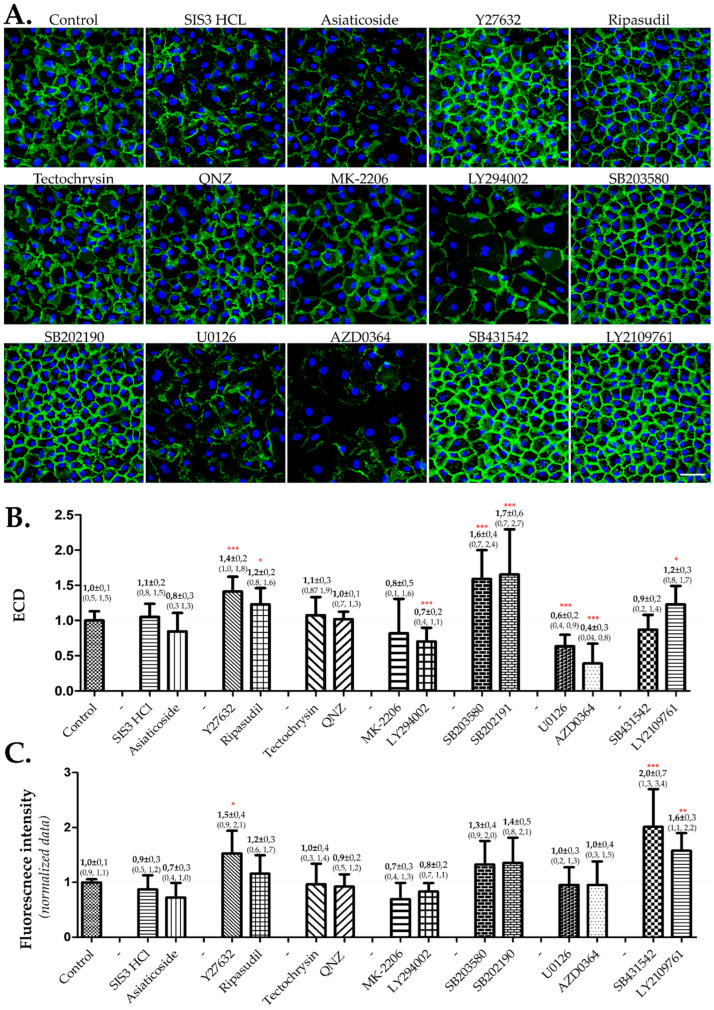
**Assessment of inhibition of different TGF β signaling pathways on hCECs cultivated in the presence of FBS.** After cell seeding in 384-well plates, the hCECs were cultured in culture medium based on BGM that contained 8% FBS for 4 weeks. BGM only (control) or BGM containing one of the inhibitors of the TGF-β signaling pathways was assessed. (**A**) Illustrative images. Cell lateral membranes were stained in green by NCAM and the nuclei were counter-stained in blue using DAPI. The images were taken in the center of each well. Objective X40, scale bar = 50 µm. (**B**) Assessment of standardized ECD. ECD was assessed by nuclei counting using DAPI staining. The ECD data were standardized by calculating the ratio with controls. At least four cell cultures from different donors were used (*n* ≥ 32 wells). (**C**) Assessment of EndMT by normalized fluorescence intensity of NAM. The IF images of NCAM were acquired with an FITC filter using the same parameters. Normalized fluorescence intensity represents the fluorescence intensity per cell/cell perimeter. The data were also standardized by calculating the ratio with the control. At least four cell cultures from different donors were used (*n* ≥ 8 wells). Significant differences by comparison to controls were denoted by red stars on the corresponding bars: *** *p* < 0.0001, ** *p* < 0.001, * *p* < 0.005.

**Figure 5 cells-12-01624-f005:**
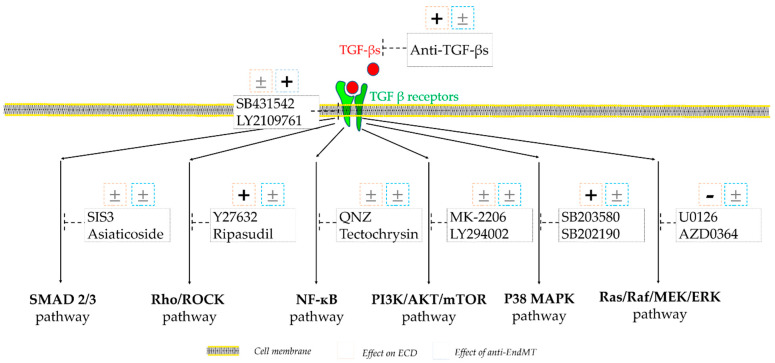
**Summary of the effects of inhibiting TGF-β signaling pathways on hCECs in vitro.** The figure illustrates the main TGF-β signaling pathways. The names of the inhibitors are indicated inside rectangles. The pink and blue squares summarize the inhibitory effects on ECD or anti-EndMT. “+” indicates a positive effect (significantly different from the control) for both tested inhibitors of the same signaling pathway, “±” indicates that at least one of the two tested inhibitors did not show a significant difference, and “−” indicates a negative effect (significantly different from control) for both inhibitors of the same signaling pathway. Appendix A. Process of analyzing the EndMT by cell shape.

**Table 1 cells-12-01624-t001:** Inhibitors of TGF-β signaling pathways.

Inhibitor	Catalog Number	Provider	Final Concentration	Signaling Pathway Inhibition	Reference
SIS3 HCL	S7959	SelleckChem	3 µM	SMAD 2/3	[30]
Asiaticoside	HY N0439	MedChemExpress	100 µg/mL	SMAD 2/3	[31]
Y27632	S1049	SelleckChem	10 µM	Rho/ROCK	[32]
Ripasudil	HY 15685	MedChemExpress	5 µM	Rho/ROCK	[33]
Tectochrysin	HY 14592	MedChemExpress	10 µg/mL	NF-κB	[34]
QNZ	S4902	SelleckChem	0.1 µM	NF-κB	[35]
MK-2206	S107811	SelleckChem	10 µM	PI3K/AKT/mTOR	[36]
LY294002	L9908	Sigma-Aldrich	20 µM	PI3K/AKT/mTOR	[37]
SB203580	S1076	SelleckChem	10 µM	P38 MAPK	[38]
SB202190	SYN-1073	SYNkinase	10 µM	P38 MAPK	[39]
U0126	6620056	CalbioChem	10 µM	Ras/Raf/MEK/ERK	[40]
AZD0364	S8708	SelleckChem	2 µM	Ras/Raf/MEK/ERK	[41]
SB431542	S1067	SelleckChem	10 µM	TGFB receptor	[28]
LY2109761	SML 2051	SelleckChem	1 µM	TGFB receptor	[27]

## Data Availability

All necessary data have been provided in this manuscript and Appendix A.

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
