# Peer review of "Investigating the Role of TGF-β Signaling Pathways in Human Corneal Endothelial Cell Primary Culture"

_cells, 2023, doi:10.3390/cells12121624_

Round 1

Reviewer 1 Report

The study entitled “Investigating the role of TGB-B signaling pathways in human corneal endothelial cell primary culture” aims to identify a biomarker for the induction of EndMT, which is a common issue with corneal cultures. In particular, the authors focus on a biomarker that is induced in repones to TGF-B signaling stemming from FBS in growth media. In the course of their work the authors identify NCAM as a marker of TGF-B induced signaling that correlated with increased EndMT as observed by cellular shape changes. In support of these findings the authors show that EndMT is inhibited if TGF-B signaling is inhibited via treatment with TGF-B function blocking antibodies. Finally, the authors attempt to pin point the transduction pathways involved and indicate a potential for the Rho/ROCK and p38 MAPK pathways as being involved. Overall, the study attempts to provide a useful and needed reagent/assay for the field however in my opinion the findings as presented appear more preliminary and lack several necessary controls. As such, I feel this study is not ready for publication in Cells in its current form.

Specific Comments:

-The organization of the manuscript is hard to follow. The methods section should simply state the protocols, and the strategy/reasoning for doing the experiments should be shifted to the results section. 

-The authors identify NCAM as the most upregulated marker in their assay, yet they compare fluorescence intensity across different antigens. This is not appropriate as each antibody will have different rates of binding and each endogenous antigen will be found in differing amounts. Comparison between treatments of the same target is appropriate, but comparison across different antigens is not informative. 

-The authors only use IF for their assay and conclusion that NCAM is a suitable target. For this to be validated the authors need to complement their IF assays with western botting and or qPCR. This is a critical verification that is needed to support the overall conclusion of the study. Similarly, the authors provide no evidence that their TGF-B 1,2 or 3 treatments are sufficient to elicit TGF-B signaling.

-it is unclear why at 10x there is clear reduction in NCAM staining with cells appearing to have little to no signal, but at 40X all cells have significant amounts of NCAM remaining at cell boundaries. Overall, there is a degree of inconsistency with the levels of NCAM between different experiments (Fig 2 vs 3). In figure 4 there appears to be fewer cells (which also appear larger) in some of the treatments. How does that affect the outcome and how was it addressed during quantification? Were these particular inhibitors toxic? Were the images collected at the same magnification?

-The authors employ many small molecule inhibitors yet there is no evidence provided that the conditions (concentration, time of exposure) at which they are being used elicit the results that are expected in primary corneal endothelial cells. Controls for the pathways being inhibited must be provided if any inference from the results is to be verified. 

-It was not made clear why the authors switch from serum free FBS to BGM media, this should be clearly addressed in the manuscript and methods. Could this change have effects on the observed outcomes?

Minor points

-Panels of Figures 1A, 2A and 3A are not referenced in the text.

-Why not also try combining TGF-B 1 + 2. 1+3 etc to see effects since FBS contains all of them?

-missing scale bar information or scale bars in figures.

-of the pathways identified to regulate EndMT, have any been previously show to affect NCAM localization or expression?

Apart from awkward organization, only minor grammatical errors and few typos detected.

Author Response

Saint-Etienne, May 23th 2023

Dear reviewer,

We have carefully considered and addressed all the points raised by your relevant review. In accordance with your comments, all our responses are specified point by point in the attached Word file and the modifications are also highlighted in yellow in the revised manuscript.

We sincerely hope our response and this revised version of the manuscript will meet your requirements and that you will find it suitable for publication.

Yours sincerely, on behalf of all the authors,

Dr. Zhiguo He

Laboratory Biology, Engineering, and Imaging for Ophthalmology,

Faculty of Medicine

Saint-Etienne

France

Reviewer 2 Report

Summary:

Corneal endothelial cell expression and function contribute to the maintenance of tissue transparency and visual acuity. Losses in its integrity and function resulting from injury or disease induce opacification and declines in visual perception. Such changes can result from pathological induced transdifferentiation  of endothelial cells into epithelial mesenchymal cells (EMT). Such rises reduce endothelial pump function to levels that are inadequate to offset the natural tendency of the stroma to imbibe fluid and become translucent. One of the current treatment approaches of endothelial dysfunction focuses on identifying novel drug targets that inhibit the EMT process and promote increases in endothelial cell proliferation. This study provides insight that may improve the outcome of corneal endothelial eye banking by suppressing EMT and inducing the reversal of endothelial cell cycle arrest. The authors need to fully evaluate the effectiveness of their drug treatments to prevent declines in endothelial cell expression and function that result from increases in the EMT process.  In addition, they should cite references identifying other  proteins in the aqueous humor that cause corneal endothelial cell expression to decline due to increases in EMT.

Major Concerns:

a) Cite more studies describing the involvement of other proteins that also induce EMT:

1)      Fibroblast growth factor-2: (FGF-2)

2)      Monocyte chemoattractant-1 protein (MCP-1)

3)      Interleukin-1β (IL-1β)

4)      Tumor necrosis factor-1α (TNF-1α)  

b) Cite Kopecny LR, Lee BWH, Coroneo MT.Ocul Surf. 2023 Jan;27:16-29. doi: 10.1016/j.jtos.2022.12.008. Epub 2022 Dec 28.PMID: 36586668, which is an excellent recent review listing references  indicating the use of ROCK inhibitors to suppress EMT.

c) It would be informative to compare the effects of different drug treatments on reversing losses in endothelial lactate efflux function resulting from increases in EMT.

d) Assess clinical applicability by evaluating the time-dependent stability of drug-induced phenotypic and functional changes suppressing EMT or increasing endothelial cell proliferation.

Minor Corrections:

Line 306: change reversed to reverse

Line 444: low concentrations of TGFβ-1 are known

Line 447: knowing not konwing

A few very minor corrections are suggested. 

Author Response

Saint-Etienne, May 23th 2023

Dear reviewer,

We have carefully considered and addressed all the points raised by your relevant review. In accordance with your comments, all our responses are specified point by point in the attached Word file and the modifications are also highlighted in green in the revised manuscript.

We sincerely hope our response and this revised version of the manuscript will meet your requirements and that you will find it suitable for publication.

Yours sincerely, on behalf of all the authors,

Dr. Zhiguo He

Laboratory Biology, Engineering, and Imaging for Ophthalmology,

Faculty of Medicine

Saint-Etienne

France

Reviewer 3 Report

This article by Aouimeur et al. studied the function of TGF-b family members in hCECs, especially the effect in EndMT. However, questions below need to be answered.

1. Is the efficacy of neutralizing antibody against different TGF-b isoforms the same? It is possible that some of the difference authors had observed are due to distinct response to the antibody. 

2. Can authors justify why they specifically chose one-way anova and Turkey HSD for statistical analysis? By looking at the graphs, the difference between some groups seem very subtle. 

3. Authors should include a line of conclusion in abstract.

Several typos are found. For example, line 293 "gourps" should be "groups". Authors need a careful check up on the spelling.

Author Response

Saint-Etienne, May 23th 2023

Dear reviewer,

We have carefully considered and addressed all the points raised by your relevant review. In accordance with your comments, all our responses are specified point by point in the attached Word file and the modifications are also highlighted in cyan in the revised manuscript.

We sincerely hope our response and this revised version of the manuscript will meet your requirements and that you will find it suitable for publication.

Yours sincerely, on behalf of all the authors,

Dr. Zhiguo He

Laboratory Biology, Engineering, and Imaging for Ophthalmology,

Faculty of Medicine

Saint-Etienne

France

Round 2

Reviewer 1 Report

I appreciate the authors efforts to improve their manuscript as well as the practical limitations. However, just because there are practical limitations this alone should not be a reason for not pursuing such experiments. In fact, the authors point out that this is critical data for the community which in my opinion should further reinforce the idea that such data needs to be fully vetted and verified as to not cause issues with subsequent studies down the line. Practically, some of the issues such as lack of sample material for qPCR is not really valid, as new kits and technologies are able to collect and analyze mRNA from very small amounts of tissue/cells. That being said, I would be supportive of publication if the following issues are addressed:

1)    Need to remove as a concluding statement from abstract any mention of molecular pathways involved (inhibitor treatment results), despite testing with inhibitors due to lack of controls for inhibitor treatments. The results can be examined in the discussion section but should not be included in the abstract as their outcomes are not properly validated. 

2)    The M&M section still contains paragraphs that read like results (section 2.5).

3)    Fig 3A- need to indicate where in the 10X image the 40X image corresponds to. There are regions, particularly in the periphery for the TGF treatments that show staining comparable to controls, while the middle of the images looks decreased. (this is also observed in other figures such as 2A) Why is this happening?

minor grammar issues

Author Response

Dear Reviewer,

We would like to extend our heartfelt appreciation for your exceptional expertise and the numerous improvements you have made to this study. We have carefully reviewed your comments and have provided our responses in the attached "Word" file. Additionally, we have incorporated all the suggested revisions into the revised manuscript, which has been included in the newly submitted version.

Thank you once again for your valuable input and for contributing to the enhancement of our research.

Respectfully,

Dr. Zhiguo He

Laboratory Biology, Engineering, and Imaging for Ophthalmology,

Faculty of Medicine

Saint-Etienne

France

Reviewer 2 Report

The authors provide effective responses to all of my concerns and suggestions for improving this significant study.  There are no other points that need to be dealt with. 

Author Response

Dear Reviewer,

We are immensely grateful for your exceptional expertise and the multitude of improvements you have made to this study. We sincerely appreciate your agreement to the publication of this study, which has considerably improved its quality and scientific rigor through your previous comments. Thank you very much for your valuable contributions.

Thank you once again for your valuable input and for contributing to the enhancement of our research.

Respectfully,

Dr. Zhiguo He

Laboratory Biology, Engineering, and Imaging for Ophthalmology,

Faculty of Medicine

Saint-Etienne

France

Round 3

Reviewer 1 Report

I am satisfied with the authors responses to my last set of concerns.